# Cheminformatics-Based Study Identifies Potential Ebola VP40 Inhibitors

**DOI:** 10.3390/ijms24076298

**Published:** 2023-03-27

**Authors:** Emmanuel Broni, Carolyn Ashley, Joseph Adams, Hammond Manu, Ebenezer Aikins, Mary Okom, Whelton A. Miller, Michael D. Wilson, Samuel K. Kwofie

**Affiliations:** 1Department of Biomedical Engineering, School of Engineering Sciences, College of Basic and Applied Sciences, University of Ghana, Legon, Accra LG 77, Ghana; 2Department of Parasitology, Noguchi Memorial Institute for Medical Research (NMIMR), College of Health Sciences (CHS), University of Ghana, Legon, Accra LG 581, Ghana; 3Department of Medicine, Loyola University Medical Center, Loyola University Chicago, Maywood, IL 60153, USA; 4Department of Molecular Pharmacology and Neuroscience, Loyola University Medical Center, Maywood, IL 60153, USA; 5Department of Chemical and Biomolecular Engineering, School of Engineering and Applied Science, University of Pennsylvania, Philadelphia, PA 19104, USA; 6Department of Biochemistry, Cell and Molecular Biology, West African Centre for Cell Biology of Infectious Pathogens, College of Basic and Applied Sciences, University of Ghana, Accra LG 54, Ghana

**Keywords:** Ebola virus, VP40, anti-Ebola, natural products, molecular docking, molecular dynamics simulations, ADMET, MM/PBSA

## Abstract

The Ebola virus (EBOV) is still highly infectious and causes severe hemorrhagic fevers in primates. However, there are no regulatorily approved drugs against the Ebola virus disease (EVD). The highly virulent and lethal nature of EVD highlights the need to develop therapeutic agents. Viral protein 40 kDa (VP40), the most abundantly expressed protein during infection, coordinates the assembly, budding, and release of viral particles into the host cell. It also regulates viral transcription and RNA replication. This study sought to identify small molecules that could potentially inhibit the VP40 protein by targeting the N-terminal domain using an in silico approach. The statistical quality of AutoDock Vina’s capacity to discriminate between inhibitors and decoys was determined, and an area under the curve of the receiver operating characteristic (AUC-ROC) curve of 0.791 was obtained. A total of 29,519 natural-product-derived compounds from Chinese and African sources as well as 2738 approved drugs were successfully screened against VP40. Using a threshold of −8 kcal/mol, a total of 7, 11, 163, and 30 compounds from the AfroDb, Northern African Natural Products Database (NANPDB), traditional Chinese medicine (TCM), and approved drugs libraries, respectively, were obtained after molecular docking. A biological activity prediction of the lead compounds suggested their potential antiviral properties. In addition, random-forest- and support-vector-machine-based algorithms predicted the compounds to be anti-Ebola with IC_50_ values in the micromolar range (less than 25 μM). A total of 42 natural-product-derived compounds were identified as potential EBOV inhibitors with desirable ADMET profiles, comprising 1, 2, and 39 compounds from NANPDB (2-hydroxyseneganolide), AfroDb (ZINC000034518176 and ZINC000095485942), and TCM, respectively. A total of 23 approved drugs, including doramectin, glecaprevir, velpatasvir, ledipasvir, avermectin B1, nafarelin acetate, danoprevir, eltrombopag, lanatoside C, and glycyrrhizin, among others, were also predicted to have potential anti-EBOV activity and can be further explored so that they may be repurposed for EVD treatment. Molecular dynamics simulations coupled with molecular mechanics Poisson–Boltzmann surface area calculations corroborated the stability and good binding affinities of the complexes (−46.97 to −118.9 kJ/mol). The potential lead compounds may have the potential to be developed as anti-EBOV drugs after experimental testing.

## 1. Introduction

Ebola virus disease (EVD) is a severely fatal disease caused by the virulent Ebola virus (EBOV). As a zoonotic infection with its putative reservoir as fruit bats [1], it has been shown that Ebola can be transmitted from animals to humans through natural means; transmission of the virus is also plausible via proximate contact with the body fluids or skin abrasions of infected people. EBOV found in semen is now a major concern as the virus can be sexually transmitted even after about 7 months from the day of recovery [2,3,4,5,6]. In general, patients infected with EVD have a predictable clinical course; thus, the early stages of the infection manifest symptoms that are considered a non-specific febrile illness, while progressing to severe gastrointestinal symptoms and severe hemorrhagic fever after weeks of infection and, subsequently, causing lethality with an up to ~90% fatality rate (with an average of about 50%) [7,8]. The virulence and lethality of this virus are due to diverse factors, particularly its ability to replicate and the assembly and budding of new virus particles at the early stages of the infection. Ebola virus is classified under the *Filoviridae* family of enveloped, single-stranded, negative-sense RNA viruses [9]. The virus forms a thread-like shape with a uniform diameter of ~80 nm and a typical length between 600 nm and 1400 nm [8,10,11].

The 2013–2016 outbreak was the largest in history, resulting in approximately 28,000 cases and 11,000 deaths [12,13]. More recently, there were EVD outbreaks in the Democratic Republic of Congo and Guinea in 2021 and Uganda in 2022. The highly virulent and lethal nature of EBOV highlights the need to develop therapeutic agents that limit the pathogenesis and spread of the virus. The genome of EBOV contains seven genes that encode seven different structural viral proteins (VP): glycoprotein (GP), nucleoprotein (NP), VP30, VP40, VP24, VP35, and RNA-dependent RNA polymerase (RdRp) [14]. Each structural protein plays a role in the lifecycle of the virus. The major matrix protein and the most extensively expressed protein, VP40, is a 326-amino acid protein that regulates viral assembly, budding, and egress and participates in transcription and viral replication through host cell RNA metabolism [15]. The X-ray structure of the VP40 protein revealed a matrix protein structure with an N-terminal domain (NTD) that is essential for oligomerization and a C- terminal shown to be essential in membrane binding through interactions with both host and cellular components [16,17].

The expression of only VP40 in mammalian cells is enough to assemble and form virus-like particles (VLPs) that are similar in size and shape to the parent virus [18]. VP40 has also been shown to regulate viral transcription, which represents a structural target in the lifecycle of the virus [19]. Crystallographic studies have shown that VP40 has two distinct domains within its structure: the NTD and the C-terminal domain (CTD) [19]. NTD interactions are responsible for VP40 dimer formation in the cytoplasm. Multiple dimers join to undergo oligomerization and form a perinuclear ring, while CTD interactions are responsible for plasma membrane binding and the coordination of higher-order oligomerization [16]. The NTD and CTD seem loosely connected, and at the NTD, the formation of ring structures (known as octameric rings) occurs where each of the eight NTD subunits can bind an RNA trinucleotide [15].

The role of the octameric ring has been identified to regulate viral transcription and replication in infected cells [19]. The crystal structure of VP40 corresponds to that of an octamer, which forms a pore-like structure and binds to RNA. The crystallographic structure of VP40 RNA shows that the Arg134 and Phe125 of VP40 mainly interact with RNA [20], and this interaction plays a critical role in octamer formation and promotes the replication of the Ebola virus.

Currently, there are two FDA-approved treatment methods for EVD. The first, REGN-EB3 (Inmazeb), is a mixture of three monoclonal antibodies that target the glycoprotein of the virus and block the attachment and entry of the virus [21,22,23]. The second, Ansuvimab (Ebanga), also prevents the virus from entering host cells, thereby limiting viral replication [24,25]. Though small molecules have been shown to have a logistical advantage over antibody therapy [26], EVD patients are limited to the use of these antibody therapies [27]. Other treatment options include supportive and symptomatic therapies, which are employed to treat the clinical symptoms of the disease [28,29]. A disproportionate number of the most advanced therapeutics currently under evaluation are small molecules directed against the RdRp required for viral replication. These include BCX4430 [30,31], favipiravir (T-705) [32], and small-interfering RNA (siRNA), the last of which suppresses viral RNA levels by degrading mRNA transcripts that interfere with translation [33].

Natural products exhibit distinctive and interesting characteristics or properties including structural and chemical diversity, thereby giving them a wider range of bioactivities. These unique properties of natural products grant them special advantages suitable for investigations regarding the discovery of new therapeutics against most infections, for which EVD is not an exception [34]. Apparently, most recent studies have emphasized the acquisition of drugs from synthetic products while being fairly neglectful of natural products. Therefore, there is a need to identify natural lead compounds from these underexploited sources that can effectively inhibit the activities of VP40 within the lifecycle of EBOV.

Following the successes of the application of computer-aided drug design (CADD) strategies to drug development processes, this study sought to virtually screen natural-product-derived compounds that target the VP40 protein and investigate the antiviral mechanisms of action of potential lead compounds. The study further profiled the potential lead compounds to test absorption, distribution, metabolism, excretion, and toxicity (ADMET). CADD approaches play key roles in drug discovery; therefore, efficient molecular docking and molecular dynamics (MD) simulations coupled with ADMET characterization will help identify potential Ebola VP40 inhibitors.

## 2. Results and Discussion

### 2.1. Protein Extraction and Preparation

A search for experimentally determined 3D structures of the EBOV VP40 protein via the Protein Data Bank (RCSB PDB) [35,36,37] revealed 13 structures. However, all 13 experimentally solved VP40 structures had numerous missing residues. For example, the structure of 1H2C, which was solved at a good resolution of 1.60 Å [20], and the structure of 7K5L, which was solved at an even better resolution of 1.38 Å [38], were both missing residues 1–68 in their NTDs. By visualizing the structures via PyMOL, it was observed that the missing residues were very close to the RNA binding site and the loop region; thus, they could not be ignored since they could influence ligand binding and may be involved in VP40–ligand interactions.

#### 2.1.1. Structure Remodeling

The amino acid sequence of the VP40 NTD extracted from UniprotKB was used to remodel the protein using Modeller and I-TASSER in order to predict a reasonable VP40 structure. A previous study used UCSF Chimera to fix the missing residues on the structure of 3TCQ prior to molecular docking [39]. Modeller generated five models using the 3D structures of 3TCQ and 7K5L as templates. All five models had a very good genetic algorithm 341 (GA341) score of 1 and DOPE scores of −1.743 × 10^4^, −1.780 × 10^4^, −1.736 × 10^4^, −1.741 × 10^4^, and −1.756 × 10^4^, respectively. However, the folding of the missing residues was not fixed in all five Modeller-generated structures. I-TASSER generated five models of the VP40 protein’s NTD with C-scores of −0.68, −1.90, −2.93, −2.73, and −1.63. The C-score determines the confidence of the structure and ranges from −5 to 2, where a higher value signifies a model with a higher confidence value and vice versa [40,41,42]. I-TASSER used a crystal structure of the VP40 with PDB ID 1ES6 as the parent template for modelling. 1ES6’s structure was experimentally determined via X-ray with a resolution of 2 Å; consequently, it was determined that only residues 1 to 43 are missing in its NTD [43]. The missing residues of all five I-TASSER structures were more reasonably folded compared to those from Modeller. Thus, the best I-TASSER model (with a C-score of −0.68 and an estimated TM-score of 0.63 ± 0.14) was selected as the most reasonable structure of VP40’s NTD (Figure 1). Aligning the selected structure to the structures of 1ES6, 7K5L, and 1H2C produced RMSD values of 0.332, 2.083, and 2.069 Å, respectively. These RMSD values are acceptable, as RMSD values below 2.5 Å are considered reasonable [44].

#### 2.1.2. Energy Minimization of Structure

The selected structure was then energy-minimized using OPLS and CHARMM36 force fields with the intent of selecting the structure with the least energy. The OPLS force field achieved the lowest energy state with −8.09 × 10^5^ kJ/mol after 845 ps (in 846 steps) while the structure with the CHARMM force field reached −7.874 × 10^5^ kJ/mol after 706 ps (in 707 steps) (Appendix A). Thus, the structure that was minimized using the OPLS force field was selected for this study.

### 2.2. Binding Site Determination

The residues Thr123, Phe125, and Arg134 of VP40 have been reported to interact with RNA [20]. Mutations of residues Lys127, Thr129, and Asn130 in the loop region of the NTD of VP40 were shown to reduce the plasma membrane localization of EBOV VP40. The mutation of these residues also significantly reduces VP40 oligomerization and limits the release of VLPs [46]. Interaction maps from previous studies show that these residues overlap with the RNA binding site [39,47]. CASTp also predicted 31 binding cavities; however, all 30 except the binding site on the loop region were relatively too small such that no compound could fit or dock into them. The binding site was predicted to have an area and volume of 303.2 Å^2^ and 438.3 Å^3^, respectively. The binding site residues consisted of Leu6, Pro7, Pro10, Met14, Ala27, Arg28, Asn31, Ser32, Asn33, Gly99, Val100, Ala101, Asp102, Lys104, Thr105, Asp144, His145, Pro146, Leu147, Arg148, Arg151, Trp191, Thr192, and Asp193. This binding site is consistent with that predicted previously using Site Finder [48]. The study also suggested that there is a higher likelihood for a small molecule to bind in this region [48].

The residues 7PTAP10 and 10PPEY13 (overlapping late (L)-budding domains: 7PTAPPEY13 in EBOV) have been experimentally shown to interact with host cellular WW-domain-bearing proteins, including neural-precursor-cell-expressed developmentally down-regulated protein 4 (Nedd4), tumor susceptibility gene 101 (Tsg101), AIP1/Alix, and HECT domain E3 ubiquitin protein ligase 1 (HECTD1), among others [49,50,51]. These interactions regulate the budding and egress of Ebola-virus-like particles (VLPs) from host cells [50,51]. Another study also showed that small molecule probes that target the PPxY domain inhibit the egress of a broad range of RNA viruses [52,53]. Trp191 stabilizes hydrophobic interactions in VP40, thus providing flexibility for the loop region to interact with lipids [54]. In vitro studies have reported that interfering with local Trp191 interactions induced dimer instability [54]. The binding site on the loop region predicted via CASTp contains these residues, thereby rendering the site a reliable target. Thus, the RNA binding site and the binding pocket predicted via CASTp (on the loop region) were considered for the molecular docking studies.

### 2.3. Validation of Docking Protocol

The ability of a docking tool to differentiate between potential binders (or inhibitors) and decoys is very important in the computational drug discovery pipeline [55,56]. AutoDock Vina was predicted to have AUC and BEDROC (α = 20) values of 0.791 (Figure 2) and 0.442, respectively. In addition, the TG and RIE values of AutoDock Vina were determined to be 0.536 and 7.315, respectively. AUC values usually range from 0 to 1, with a value of 1 implying a perfect degree of classification between actives and decoys. AUC values within the 0.5 to 0.7 range are considered to provide moderate distinctive ability, while values below 0.5 imply poor discrimination ability between active and decoys [57,58]. However, a molecular docking method with an AUC above 0.5 and a TG above 0.4 is considered good performance and is reproducible under similar experimental conditions [59]. The results presented herein show that AutoDock Vina performed very well and can reasonably distinguish between VP40 actives and inactives.

### 2.4. Molecular Docking Studies

The AfroDb, NANPDB, and TCM libraries were obtained and virtually screened against VP40 to shortlist compounds with reasonably good docking scores. The docking results were ranked according to docking scores in a decreasing order. A docking score threshold of −8 kcal/mol and lower was used to filter the top compounds from the AfroDb, NANPDB, TCM, and Approved compounds libraries since nilotinib (a known inhibitor with the best docking score) had a docking score of −7.9 kcal/mol. 

For the AfroDb library, 773 compounds were successfully screened against the VP40 protein. Only 7 AfroDb compounds had docking scores of −8 kcal/mol and below. The lowest docking score was observed for ZINC000095486217 with a score of −8.6 kcal/mol, while ZINC000028462577 had a score of −8.3 kcal/mol. The compounds ZINC000034518176 and ZINC000095485942 both had a docking score of −8.1 kcal/mol (Table 1), while ZINC000014825190, ZINC000085594516, and ZINC000095486263 all had a docking score of −8 kcal/mol.

For the NANPDB library, a total of 3615 compounds were successfully screened against the VP40 protein. A total of 11 compounds were observed to have docking scores of −8 kcal/mol and below. The compound NANPDB4060 demonstrated the lowest docking score with respect to VP40 with a value of −8.8 kcal/mol, followed by NANPDB2933 (Table 1) and NANPDB4059 with docking scores of −8.5 and −8.4 kcal/mol, respectively. NANPDB322, NANPDB362, NANPDB4055, and NANPDB5641 all had docking scores of −8.1 kcal/mol, while NANPDB4057, NANPDB4240, NANPDB552, and NANPDB6298 all had a docking score of −8 kcal/mol.

For the TCM library, a total of 25,131 compounds were successfully screened against VP40. The docking scores of all 25,131 compounds in the TCM library ranged from −1.4 to −9 kcal/mol. A total of 163 compounds were observed to have docking scores of −8 kcal/mol and lower. ZINC000103579839 demonstrated the lowest docking score with respect to VP40 (at −9 kcal/mol) among the TCM compounds, followed by ZINC000085531689 with a score of −8.9 kcal/mol. The compounds ZINC000014089759, ZINC000085545967, ZINC000085568633, ZINC000085593149, and ZINC000085991498 all had a docking score of −8.8 kcal/mol.

A total of 2738 approved drugs were successfully screened against the VP40 protein, of which 30 compounds had docking scores of −8 kcal/mol and lower. Doramectin had the lowest docking score of −9.1 kcal/mol, followed by ledipasvir with a docking score of −9 kcal/mol (Table 1). Avermectin B1 (abamectin) and elbasvir both had a docking score of −8.7, while venetoclax (ABT-199 or GDC-0199) and revefenacin both had a docking score of −8.5 kcal/mol (Table 1). Glecaprevir also had a docking score of −8.3 kcal/mol (Table 1). Doramectin and abamectin are avermectins, a class of macrocyclic lactone compounds that are widely used in the veterinary field for the treatment of parasites [60,61]. Furthermore, the antiviral activity of avermectins such as ivermectin and abamectin have been recorded in literature [62,63,64,65,66,67]. Abamectin and ivermectin were reported to inhibit Chikungunya virus (CHIKV) with EC_50_ values of 1.5 and 0.6 μM, respectively [62]. They were also reported to have antiviral effects on alphaviruses (Semliki Forest and Sindbis virus) and yellow fever virus [62]. Yellow fever virus is a flavivirus, a small enveloped RNA virus, and Ebola is also an RNA virus [68,69]. Although flaviviruses and filoviruses have different modes of transmission and symptoms, they may both be characterized by viral hemorrhagic fevers (VHFs) [68,69,70]. Ledipasvir and elbasvir are hepatitis C virus (HCV) NS5A inhibitors that help prevent viral RNA replication, assembly, and the viral release of HCV infection [71,72]. Glecaprevir is also an HCV NS3/4A protease inhibitor [73,74,75], making it an interesting candidate to test and repurpose for Ebola treatment. Venetoclax, a B-cell lymphoma 2 inhibitor [76,77] used in chronic lymphocytic leukemia and small lymphocytic lymphoma treatments [78,79], was shown to reduce latent virus in a T cell model when used in combination with ixazomib [80]. There have been cases where Ebola still persists in immune-privileged regions (including the eyes and testes) in survivors [81]. Experimental testing is required to determine if venetoclax could help reduce the viral load in immune-privileged sites after EVD treatment regimens are completed.

For the curated inhibitors, the docking scores ranged from −4.4 to −7.9 kcal/mol for the 43 successfully docked compounds. Nilotinib demonstrated the lowest docking score of −7.9 kcal/mol, followed by imatinib, cepharanthine, and daunomycin with docking scores of −7.6, −7.3, and −6.9 kcal/mol, respectively. Azarclorzine and mefloquine both had a docking score of −6.6 kcal/mol, while bosutinib, mebendazole, raloxifene, sangivamycin, sunitinib, and topotecan all had a docking score of −6.3 kcal/mol. Nilotinib and imatinib, which were used as standards in this study, have previously been shown to reduce VP40 in VLPs at the concentrations tested (10 and 20 μM) with insignificant toxicities [82].

An in silico study used AutoDock 4.2 to virtually screen five compounds (vindesine, BIX-01294, NVP-ADW742, ZINC91973695, and ZINC67869167) against VP40 [39]. These five compounds were experimentally shown to inhibit EBOV [39]. The docking scores of the five compounds ranged from −4 to −5 kcal/mol [39]. The docking score rankings of the five compounds from AutoDock 4.2 were consistent with their experimental half maximal inhibitory concentration (IC_50_) rankings [39,83]. Another study also screened compounds in the MCULE database against VP40 using AutoDock Vina and the docking scores of the top compounds ranged from −5.9 to −7 kcal/mol [47]. Herein, the shortlisted compounds demonstrated better docking scores than those of previous studies [39,47]. The relatively better docking scores of the compounds reported herein make them interesting candidates for further investigation into their potential EBOV-VP40-binding and inhibition properties.

### 2.5. Absorption, Distribution, Metabolism, Excretion, and Toxicity (ADMET) Profiles of the Shortlisted Compounds

ADMET prediction was further performed on shortlisted compounds from the AfroDb, NANPDB, and TCM libraries to analyze their pharmacological profiles. It is essential for a parent compound to have an impressive ADMET profile. Lipinski’s rule of five and Veber’s rule were applied in the prediction of drug-like and physiochemical properties of the shortlisted compounds. These rules are a set of property values that were derived from the classification of the key physicochemical properties of drug-like compounds [84]. According to Lipinski’s rule, a drug-like compound should have a molecular weight ≤500 Daltons; a logP value ≤5; a number of hydrogen bond donors ≤5; and a number of hydrogen bond acceptors ≤10 [85,86,87]. According to Lipinski, a compound is drug-like when it is within the acceptable range of the rule of five or when it violates only one of the rules. On the other hand, Veber’s rule dictates that a good orally bioavailable drug should possesses 10 or fewer rotatable bonds and a topological polar surface area (TPSA) less than 140 Å^2^ [88].

Based on Lipinski’s rule, the compounds ZINC000095486217, ZINC000085594516, and ZINC000095486263 from the AfroDb library had two, two, and three violations, respectively, and were thus eliminated from further analyses. ZINC000028462577 was also predicted to violate Veber’s rule, as it had a TPSA of 170.8 Å^2^. Only three AfroDb compounds, namely, ZINC000034518176, ZINC000095485942, and ZINC000014825190, were in accordance with both Lipinski’s and Veber’s rules. For the NANPDB library, only NANPDB2933 was in accordance with both Lipinski’s and Veber’s rules (Table 2). NANPDB2933 has a molecular weight of 486.51 g/mol, a TPSA of 132.5 Å^2^, and a consensus logP of 1.6 (Table 2). A total of seven and ten compounds violated only Lipinski’s and only Veber’s rules, respectively, while seven violated both Lipinski’s and Veber’s rules. A total of 84 compounds out of the 163 TCM compounds with favorable docking scores complied with both Lipinski’s and Veber’s rules. The ADMET results of the shortlisted compounds are summarized in Table 2 and Appendix A.

The compounds were further subjected to toxicity testing using OSIRIS DataWarrior 5.5.0 [89]. ZINC000014825190 (AfroDb) was predicted to have low tumorigenic and high reproductive effects and was thus eliminated. ZINC000034518176, ZINC000095485942, and NANPDB2933 were predicted to be non-tumorigenic, non-mutagenic, and non-irritants and had no reproductive effects. For the TCM library, a total of 41 out of the 84 top compounds were predicted to be less toxic, with no mutagenic, tumorigenic, irritancy-inducing, or reproductive effects. ZINC000085568633, with a docking score of −8.8 kcal/mol, was predicted to be highly tumorigenic, while ZINC000085991498 (which also had a docking score of −8.8 kcal/mol) was predicted to have low-level mutagenic effects. In all, 43 compounds from the 3 natural product libraries (1 NANPDB, 2 AfroDb, and 41 TCM compounds) were shortlisted as potentially safe binders of the VP40 NTD. The ADMET predictions indicate that the shortlisted compounds have desirable ADMET properties with insignificant toxicity. 

### 2.6. Protein–Ligand Interactions of Lead Compounds

Visualizing the protein–ligand complexes of the top compounds in PyMOL showed that all the shortlisted ligands had docked deep into the active sites. Further analysis was carried out to identify the amino acid residues of the VP40 protein that interact with the ligands by using Ligplot+ to visualize the interactions (Figure 3 and Appendix A–j). ZINC000034518176 docked at the RNA binding site, interacted with Gly139 (with a bond length of 3.05 Å) via hydrogen bonds, and formed hydrophobic bonds with Phe36, Asn43, Pro47, Thr121, Phe125, Arg134, Asn136, Arg137, and Leu138 (Table 1 and Appendix A). The other shortlisted compounds were observed to bind in the loop region. It was observed that ZINC000085568136, ZINC000070454124, ledipasvir, elbasvir, revefenacin, nilotinib, and imatinib docked in the same site on the loop region (Table 1). Nilotinib did not form any hydrogen bonds with VP40 (Table 1). However, nine hydrophobic interactions were observed for the VP40–nilotinib complex, namely, with the Tyr18, Arg21, Pro39, Val42, Gly44, Gly126, Lys127, Ala128, and Thr129 residues (Figure 3a and Table 1).

Most of the top compounds were observed to dock in the loop region surrounded by the Lys127, Ala128, Thr129, Asn130, Pro131, Gln159, Glu160, Pro164, Pro165, Val166, Gln167, Leu168, Pro169, and Gln170 residues. ZINC000085531689, with a docking score of −8.9 kcal/mol, formed hydrogen bonds with Asn130 (2.99 and 3.05 Å), Gln159 (2.94 and 3.22 Å), and Pro169 (2.71 Å) and interacted with the Glu160, Gln167, Leu168, Gln170, and Phe172 residues via hydrophobic bonds (Table 1 and Figure 3b). According to the interaction maps, doramectin, which had the lowest docking score with respect to VP40 (−9.1 kcal/mol), also formed five hydrogen bonds with Glu12, Asn130, Gln159, Leu168, and Pro169 with bond lengths of 2.72, 3.14, 2.8, 3.33, and 3.06 Å, respectively (Table 1 and Appendix A). Doramectin also interacted via hydrophobic bonding with Tyr13, Lys127, Ala128, Thr129, Pro131, Glu160, Phe161, Pro165, and Gln170 (Table 1 and Appendix A). The overlapping L-budding domains consisting of residues 7PTAPPEY13 have been experimentally reported to interact with host cellular WW domain proteins including Nedd4, Tsg101, AIP1/Alix, and HECTD1 [49,50,51]. Interactions with these residues regulate the budding and egress of Ebola-virus-like particles (VLPs) from host cells [50,51].

The compounds ZINC000014089759 and ZINC000014089743 were also observed to bind in a different area on the loop region, forming hydrophobic interactions with Tyr13, Tyr18, Pro19, Asn23, and Leu127 (Table 1). ZINC000014089759 also formed hydrogen bonds with Arg21 (3.02, 3.03 Å), Pro39 (3.1 Å), and Lys127 (3.04 Å) (Appendix A) while ZINC000014089743 interacted via hydrogen bonds with Arg21 (2.96 Å), Ser24 (2.97 Å), and Pro39 (3.09 Å). In the loop region, Lys127 was observed to be involved in the VP40–ligand interaction.

A recent in silico study reported that the topmost compound interacted with the following residues: Thr123, His124, Phe125, Gly126, Lys127, Arg134, and Tyr171 [47]. In another study, vindesine was predicted to form hydrogen bonds with Gln35, Gln38, and Lys127 [39]. BIX-01294 was also shown to interact with Gln33 and Ile34, while NVP-ADW742 interacted with Lys127 via a hydrogen bond. Lys127 was suggested to play a critical role in ligand binding [39].

### 2.7. Prediction of Biological Activities of Lead Compounds

Even though molecular docking studies can provide valuable insights into the poses and interactions between molecules and their targets, without experimental validation, it is difficult to determine the true accuracy of the predictions. In the midst of this lacuna, this study employed additional computational approaches to mitigate the potential errors in the molecular docking process. In this study, the biological activity of the shortlisted compounds was predicted using a Bayesian algorithm [90,91,92,93]. The anti-EBOV inhibition efficiencies of the compounds were also predicted using RF and SVM models [94]. Structural similarity searches of the shortlisted compounds were also performed to identify compounds with known antiviral or anti-EBOV related activity. Structural features are often critical determinants of biological activity, and, in many cases, compounds with structural features similar to known active compounds can exhibit similar biological activity [95,96].

#### 2.7.1. PASS Predictions

The biological activity spectrum (BAS) of a compound is a characteristic that reflects the compound’s various pharmacological effects and its physiological and biochemical mechanisms of action [90]. A BAS also accounts for specific toxicities such as mutagenicity, carcinogenicity, teratogenicity, and embryo toxicity [90]. This prediction operates on the principle that the biological activity of a compound equates to its structure; therefore, this activity is largely dependent on the structural nature of a compound. The majority of biologically active compounds have both pharmacotherapeutic and side-effect-inducing/toxic actions. The biological activity of a compound gives insight into its mechanism of action against a therapeutic target.

In this study, PASS was employed to predict the biological activities of the analyzed potential lead compounds [90,91,92,93]. The compounds ZINC000034518176, ZINC000085531689, ZINC000014089759, ZINC000085545967, ZINC000014089743, and ZINC000101564200 were predicted to be antivirals with anti-influenza activity, exhibiting Pa values of 0.793, 0.214, 0.739, 0.243, 0.241, and 0.356 and Pi values of 0.003, 0.176, 0.004, 0.140, 0.115, and 0.062, respectively. The compounds were also predicted to possess other antiviral activities, including against rhinovirus, herpes, HIV, and hepatitis B, with their Pa values greater than their Pi values. ZINC000101564200 and ZINC000014089743 were also predicted to be human coronavirus 3C-like protease inhibitors with Pa values of 0.270 and 0.249 and Pi values of 0.050 and 0.078, respectively. ZINC000101564200 was further predicted to be an inhibitor of simian immunodeficiency virus proteinase (Pa: 0.318 and Pi: 0.105) and HIV-1 integrase and possesses antiviral properties against adenoviruses (Pa: 0.391; Pi: 0.033), picornaviruses (Pa: 0.353; Pi: 0.155), cytomegalovirus (CMV) (Pa: 0.243; Pi: 0.087), and poxviruses (Pa: 0.216; Pi: 0.135).

Furthermore, NANPDB2933, ZINC000085531689, ZINC000014089759, ZINC000085545967, ZINC000014089743, and ZINC000101564200 were predicted to be RNA synthesis inhibitors. Favipiravir, a selective RdRp (RNA polymerase) inhibitor, exhibits activity against single-stranded RNA viruses (including EBOV) by inhibiting viral replication [2,97,98,99]. ZINC000034518176 and ZINC000101564200 were also predicted to be viral entry inhibitors. Enfuvirtide, a peptide used for HIV treatment, is a viral entry inhibitor [100,101]. LJ001 is a viral entry small molecule inhibitor that has been shown to be effective against a wide range of enveloped viruses [102]. Enveloped viruses such as EBOV, HIV, and coronaviruses have been reported to have similar cell fusion mechanisms [100]. These predictions provide evidence for the antiviral properties of the shortlisted compounds, rendering them interesting candidates for further in vitro probing.

NANPDB2933, ZINC000085531689, ZINC000014089759, and ZINC000085545967 were predicted to be antihemorrhagic. Antihemorrhagic agents may help reduce the risk of bleeding and improve survival rates in patients with severe Ebola infection. ZINC000034518176, ZINC000095485942, NANPDB2933, ZINC000014089759, ZINC000085545967, ZINC000014089743, and ZINC000101564200 were also predicted to be hepatoprotectants with Pa values of 0.926, 0.377, 0.282, 0.929, 0.429, 0.932, and 0.317 and Pi values of 0.002, 0.036, 0.066, 0.002, 0.027, 0.002, and 0.054, respectively. Liver damage is one of the hallmarks of EVD infection [103,104,105]. These compounds may be beneficial with respect to managing liver failure and may support the liver during recovery from EVD.

#### 2.7.2. Structural Similarity Search

A structural similarity search of the potential lead compounds via DrugBank [106,107] revealed that ZINC000034518176 is structurally similar to ginsenosides and beta-sitosterol with scores of 0.705 and 0.785, respectively. Ginsenosides, the main active ingredients in ginseng, are widely known for their antiviral properties [108,109,110,111,112]. β-sitosterol (50 mg/kg), the main component of *Pinellia ternata* (Thunb.) Breit., was reported to inhibit ~95.79% of white spot syndrome virus (WSSV) in crayfish [113]. In addition, the stem bark, root, and leaf extracts of *Erythrostemon yucatanensis* (Greenm.) Gagnon and GP Lewis were reported to have anti-influenza-virus activity [114]. Analysis of spectroscopy data revealed that a combination of three phytosterols, namely, β-sitosterol, stigmasterol, and campesterol, in the stem bark active fraction was responsible for attenuating hemagglutinin binding and affected viral particle infectivity (IC_50_ of 3.125 µg/mL) [114].

ZINC000034518176, ZINC000014089759 (11-keto boswellic acid), and ZINC000014089743 are also structurally similar to ursolic acid, presenting similarity scores of 0.736, 0.761, and 0.882, respectively. Ursolic acid has been shown to possess broad spectrum antiviral activity [115,116]. Ursolic acid (10 µM) was reported to inhibit rotavirus replication with no cytotoxicity in MA104 cells [117]. Ursolic acid, an extract of the Chinese herb *Fructus Ligustri Lucidi*, was reported to inhibit HCV (JFH1) at an IC_50_ of 10.6 μg/mL [118].

ZINC000014089743 was predicted to be structurally similar to madecassic acid (0.863), asiatic acid (0.863), enoxolone (0.807), and betulinic acid (0.701). Madecassic acid and asiatic acid (up to 10 μM) showed moderate and weak inhibitory activities, respectively, against HIV-1 viral protein R [119]. Enoxolone (glycyrrhetinic acid (GA)) is a hepatoprotective agent that inhibited the activation of hepatic inflammation in a hepatitis-virus-infected mouse [120]. Four derivatives of GA were shown to inhibit hepatitis B DNA replication with IC_50_ values less than 10 μM while GA had an IC_50_ of 39.28 μM [121]. GA’s anti-SARS-CoV-2 [122,123], anti-influenza [124], and anti-Epstein–Barr virus [125] activities have been reported previously. Betulinic acid has been reported to possess antiviral properties. It was reported to inhibit dengue virus 2 (DENV2) in Huh7, HepG2, HEK293T, BHK-21, and Vero cells with IC_50_ values of 0.9463, 0.8038, 0.9463, 0.7697, and 3.224 μM, respectively [126]. Betulinic acid (IC_50_ < 2 μM) also inhibited three other serotypes of DENV: DENV1, DENV3, and DENV4 [126]. The inhibition of ZIKV (in JEG-3 cells) and CHIKV (in SJCRH30 cells) with IC_50_ values of 2.45 and 0.6853 μM, respectively, was also reported [126]. The anti-influenza [127], anti-SARS-CoV-2 [128], anti-hepatitis [129,130] anti-HIV [131,132], and anti-herpetic [133] activities of betulinic acid have also been highlighted in the literature.

ZINC000101564200 is structurally similar to anthralin (0.754) and emodin (0.711). Anthralin suppresses influenza A virus (IAV) proliferation [134] and has been shown to be an HIV-latency-reversing agent [135]. Therefore, ZINC000101564200 may be useful for EBOV latency reversal to completely eradicate EBOV from patients’ bodies. Emodin has been reported to have virucidal effects on Zika virus (ZIKV) in which it provokes a reduction in viral load of 83.3% at 40 µM [136]. Emodin also possesses anti-IAV [137,138,139], anti-CMV [140], and anti-HBV [141] properties. The results from our structural similarity search corroborate the likelihood of the shortlisted compounds being potential antivirals.

#### 2.7.3. Anti-Ebola Activity Prediction

The anti-EBOV inhibition efficiencies of the shortlisted compounds were predicted via RF and SVM models using Anti-Ebola [94]. Anti-Ebola is a regression-based prediction algorithm that predicts the potential EBOV-inhibitory activity of a query compound using quantitative structure–activity relationship (QSAR) analysis [94]. These models were previously validated using compounds that have been experimentally shown to possess anti-EBOV activity [94]. The model predicted that indinavir (0.03 μM), maraviroc (0.30 μM), abacavir (1.27 μM), tilorone (1.95 μM), pyronaridine (0.5 μM), and quinacrine (2 nM) were anti-EBOV compounds [94], and these results are consistent with experimental findings [142,143].

In this study, the compounds ZINC000034518176 and ZINC000095485942 from the AfroDb library were predicted by the RF model of Anti-Ebola [94] to have IC_50_ values of 3.86 and 2.93 μM, respectively, while the SVM model predicted IC_50_ values of 3.14 and 20.15 μM, respectively (Appendix A). Both models predicted that NANPDB2933 had an IC_50_ of 2 μM (Appendix A). The compounds whose inhibition efficacy could not be predicted were automatically eliminated. Anti-Ebola was not able to predict the inhibitory efficiency of ZINC000085504890 and ZINC000070451048 from the TCM library. In addition, the Ebola inhibition efficiencies of venetoclax, hederacoside C, pibrentasvir, and efonidipine (approved drugs library) could not be predicted. However, revefenacin, BMS-927711 (BHV-3000), and irinotecan (CPT-11) were predicted to show no anti-Ebola activity [IC_50_ of 1 × 10^6^ μM (1 × 10^3^ mM or 1 M)] by both the RF and SVM models and were thus removed from the shortlisted compounds (Appendix A). Irinotecan, a DNA topoisomerase I inhibitor, was previously reported to have no antiviral effect on pseudorabies virus (PRV) infection [144], thereby lending support to the prediction that these compounds likely do not have any anti-Ebola properties.

Sennoside A was recently shown to inhibit Zika virus protease with IC_50_ and Ki values of 0.66 and 0.19 μM, respectively [145]. Sennoside A and B also inhibited HIV-1 ribonuclease H and reverse-transcriptase-associated DNA polymerase activities with IC_50_ values in the 2–5 µM range [146]. In this study, sennoside A and B, which are both anthraquinone glycosides [147], were predicted to be potential anti-EBOV compounds with IC_50_ values below the micromolar range (Appendix A). Similarly, ledipasvir, avermectin B1, nafarelin acetate, danoprevir (ITMN-191), eltrombopag, lanatoside C, glycyrrhizin (glycyrrhizic acid), and daclatasvir digydrochloride were also predicted to possess anti-EBOV properties in very low micromolar ranges (Appendix A). The broad spectrum antiviral activity of lanatoside C has been reported [148]. Lantoside C was identified as being able to effectively inhibit all four serotypes of dengue (with an IC_50_ of 0.19 μM in HuH-7 cells), Kunjin, Chikungunya, and Sindbis and the human enterovirus 71 viruses [148]. Like ledipasvir, danoprevir is a known directly acting antiviral used in the treatment of chronic HCV infection [149,150,151]. Eltrombopag is known to increase platelet counts in HCV-associated thrombocytopenia [152,153] and can be beneficial in the management of bleeding in cases of Ebola, a viral hemorrhagic fever. Some EVD patients have been reported to experience thrombocytopenia [154,155]. Glycyrrhizin’s wide range of antiviral properties has been documented in literature [156,157,158,159,160]. These predictions and the antiviral-related properties of these compounds make them interesting candidates for experimentally probing to determine their anti-EBOV activity and selectivity.

### 2.8. Molecular Dynamics Simulations of Some Selected Compounds

MD simulations are used not only to explore the conformational aspects of biological systems but also to obtain a significant scope with which to analyze protein–ligand interactions. A 100 ns MD simulation was performed to investigate the dynamic behavior of the lead complexes. The simulation was conducted for the unbound protein (VP40), nilotinib, and seven potential lead compounds complexed with VP40.

#### 2.8.1. Root-Mean-Squared Deviation (RMSD)

To explore the intricate interactions between VP40 and the analyzed ligands and investigate the stability of the lead complexes, the RMSD was analyzed for the 100 ns MD simulation. The VP40–ZINC000034518176 complex demonstrated the greatest stability among all the systems, presenting an average RMSD of 0.33 ± 0.039 nm. The unbound VP40 protein had an average RMSD of 0.37 ± 0.048 nm. VP40 in complex with nilotinib, ZINC000085568136, ZINC000085531689, ZINC000095912717, and ZINC000014089759 presented average RMSD values of 0.404 ± 0.051, 0.45 ± 0.077, 0.39 ± 0.057, 0.37 ± 0.045, and 0.37 ± 0.046 nm, respectively. All the systems were observed to reach equilibrium around 25–35 ns (Figure 4). The VP40–ZINC000085568136 complex demonstrated the highest RMSD, which reached 0.5 nm around 40 ns and was maintained until the end of the 100 ns simulation period (Figure 4). Similar RMSD values (~0.45 to 0.48 nm) were reported previously for the top two compounds (emodin-8-beta-D-glucoside and tonkinochromane G) identified via virtual screening and MD simulations of VP40 [9]. The relatively higher RMSD values obtained for VP40 are not surprising, as loop regions or proteins with more loops tend to have higher RMSD values than alpha helices and beta sheets [161].

#### 2.8.2. Root Mean Square Fluctuations (RMSF)

The RMSF of each complex was analyzed to investigate each residue’s average fluctuation throughout the 100 ns simulation (Appendix A). The RMSF provides information on residues that are involved in ligand binding, as these residues tend to have lower fluctuations due to their interactions with the ligand that stabilizes the complex. In many cases, binding interactions can stabilize the region and reduce the flexibility of residues involved in binding. For all the VP40–ligand complexes, fluctuations were observed at similar regions of VP40 (Appendix A). Major fluctuations were observed at 1–50, 60–72, and 125–131, while minor fluctuations were observed at residue indexes of 80–94, 101–110, and 140–145 (Appendix A). The highest RMSF was observed for the VP40–NANPDB2933 complex at residue index 29–36 with values ranging from 0.5826 nm (Gly35) to 0.9188 nm (Asn31). The Gly29, Gly30, Ser32, Asn33, Thr34, and Phe36 residues had RMSF values of 0.8307, 0.8799, 0.804, 0.7405, 0.6474, and 0.7089 nm, respectively, for the VP40–NANPDB2933 complex (Appendix A). Positions 72–80, 95–100, 119–122, and 172–183 demonstrated the lowest fluctuations (Appendix A). These residue indexes could be involved in ligand binding and warrant further investigation.

#### 2.8.3. Radius of Gyration (Rg)

The compactness of the complexes was assessed by analyzing the radius of gyration (Figure 5). A steady radius of gyration throughout a simulation period indicates a stably folded protein [162]. The Rg for the unbound protein was relatively steady over the simulation period, corresponding to an average of 1.67 ± 0.01 nm (Figure 5). All the VP40–ligand complexes had Rg values comparable to that of the unbound protein. The VP40–nilotinib, VP40–ZINC000014089759, VP40–ZINC000034518176, and VP40–ZINC000095485942 complexes had average Rg values lower than that of the unbound VP40, presenting values of 1.668 ± 0.025, 1.653 ± 0.01, 1.638 ± 0.018, and 1.635 ± 0.012 nm, respectively (Figure 5). The VP40–ZINC000085568136, VP40–ZINC000085531689, VP40–ZINC000095912717, and VP40–NANPDB2933 complexes exhibited comparable Rg scores, presenting average values of 1.683 ± 0.015, 1.685 ± 0.016, 1.685 ± 0.01, and 1.681 ± 0.023 nm, respectively (Figure 5).

#### 2.8.4. Hydrogen Bond Analysis

The hydrogen bonds formed between VP40 and the studied ligands were analyzed using “gmx hbond”. The number of hydrogen bonds formed at every nanosecond was determined. Snapshots taken at 25 ns intervals (0, 25, 50, 75, and 100 ns) for each system during the 100 ns MD simulations were also generated, and the interaction profiles were determined using LigPlot+ (v1.4.5) (Appendix A). On average, the VP40–nilotinib, VP40–ZINC000095485942, VP40–NANPDB2933, VP40–ZINC000085531689, VP40–ZINC000014089759, VP40–ZINC000085568136, and VP40–ZINC000095912717 complexes had 0.634 ± 0.784, 0.257 ± 0.56, 2.693 ± 1.046, 1.446 ± 1.22, 0.139 ± 0.347, 0.782 ± 0.701, and 1.337 ± 0.886 hydrogen bonds, respectively, throughout the simulation period. ZINC000034518176, on the other hand, was observed to form no hydrogen bonds with VP40 throughout the 100 ns period. However, the interaction maps showed that ZINC000034518176 formed only one hydrogen bond with Gly139 (bond length of 3.02 Å) at the 0 ns timeframe but could not maintain this interaction throughout the simulation period (Appendix A).

According to “gmx hbond”, the highest number of hydrogen bonds (five H bonds) formed were observed for VP40–ZINC000085531689 (at time = 0 ns) and NANPDB2933 (at 23 ns and 66 ns). However, the interaction maps revealed that ZINC000085531689 formed six hydrogen bonds with Asn130 (2.79, 3.16, and 3.24 Å), Gln159 (3.02 Å), Glu160 (2.53 Å), and Pro169 (2.63 Å) and interacted via hydrophobic interactions with Gln167, Leu168, Gln170, and Phe172 at 0 ns (Appendix A). NANPDB2933 interacted with His124 (2.91 Å), Val166 (3.03 and 3.28 Å), and Leu168 (3.25 Å) via hydrogen bonds and formed hydrophobic bonds with Met89, Ala128, Thr129, Asn130, Pro131, Glu160, Pro164, Pro165, Gln167, and Phe172 at 25 ns. At 50 ns, hydrogen bond interactions with Val166 (2.88 and 3.25), Leu168 (3.2 and 3.34 Å), and Gln170 (2.85 Å) were observed (Appendix A). 

At the end of the simulation, “gmx hbond” showed that NANPDB2933, ZINC000095485942, and ZINC000085531689 formed three, one, and one hydrogen bonds with VP40, respectively, while the other ligands lost their hydrogen bonds with VP40 (Appendix A). According to the interaction profile, at the end of the simulation (100 ns), NANPDB2933 formed four hydrogen bond interactions with Asn130 (3.05 Å), Val166 (2.79 and 3.31 Å), and Gln170 (2.73 Å) and formed hydrophobic interactions with His124, Ala128, Thr129, Pro131, Val133, Glu160, Leu163, Leu168, and Phe172 (Appendix A). The hydrogen bonds formed between NANPDB2933 and VP40 could influence the complex’s binding properties and stability and the potential activity of NANPDB2933 since multiple hydrogen bonds have been reported to contribute to ligand activity [163]. In addition, ZINC000095485942 formed two hydrogen bonds with Met1 (2.52 and 2.78 Å), while ZINC000085531689 also formed two hydrogen bonds with Glu160 (2.95 Å) and Gln167 (3.32 Å) at 100 ns (Appendix A).

### 2.9. Evaluation of Lead Compounds Using MM/PBSA Computations

#### 2.9.1. Contributing Energy Terms

MM/PBSA calculations were performed to evaluate the binding free energies of the VP40–ligand complexes after the MD simulation [164,165]. The strength of the interaction between a protein and a ligand is quantified by its binding free energy. The known inhibitor, nilotinib, had a binding free energy of −11.21 kJ/mol according to the MM/PBSA calculations (Table 3). All the potential lead compounds demonstrated lower binding free energies than nilotinib, implying that they may have higher binding affinity for VP40 than nilotinib (Table 3). The binding free energy values of the VP40–potential lead complexes ranged from −46.97 kJ/mol to −118.9 kJ/mol, with NANPDB2933 showing the highest affinity to VP40 (Table 3). The very high binding affinity demonstrated by NANPDB2933 is not surprising, as it demonstrated the most developed hydrogen bond network with VP40 (Appendix A). The VP40–ZINC000034518176, VP40–ZINC000095485942, VP40–ZINC000085568136, VP40–ZINC000014089759, VP40–ZINC000095912717, and VP40–ZINC000085531689 complexes had binding free energies of −105, −103.3, −92.71, −70.52, −66.99, and −46.97 kJ/mol, respectively (Table 3). The relatively low standard deviation values of the binding free energies (<4 kJ/mol) signify the reliability of the simulations (Table 3).

#### 2.9.2. Energy Decomposition per Residue

The energies contributed by each residue were computed using MM/PBSA calculations to show the key residues that are involved in VP40–ligand binding (Figure 6 and Appendix A) [164,166]. Residues with energy contribution values <−5.0 or >5.0 kJ/mol are critical in the binding of a ligand to a protein [167]. Residues contributing <−5.0 kJ/mol improve ligand binding while those with energy levels >5.0 kJ/mol impair binding potency. For the VP40–nilotinib complex, Met1 (20.53 kJ/mol), Arg2 (22.31 kJ/mol), Arg3 (18.97 kJ/mol), Arg21 (43.2 kJ/mol), Arg28 (26.68 kJ/mol), Arg52 (21.38 kJ/mol), His64 (16.91 kJ/mol), Lys86 (20.28 kJ/mol), Lys90 (18.33 kJ/mol), Lys104 (15.86 kJ/mol), Lys127 (42.41 kJ/mol), Arg134 (41.27 kJ/mol), Arg137 (24.97 kJ/mol), Arg148 (25.69 kJ/mol), Arg151 (28.08 kJ/mol), and Lys180 (18.36 kJ/mol) were observed to contribute unfavorably to ligand binding (Appendix A). On the other hand, Glu12 (−27.69 kJ/mol), Glu15 (−29.69 kJ/mol), Glu40 (−43.26 kJ/mol), Val42 (−9.368 kJ/mol), Asp45 (−37.24 kJ/mol), Asp56 (−17.77 kJ/mol), Asp57 (−15.92 kJ/mol), Asp60 (−15.25 kJ/mol), Glu76 (−19.97 kJ/mol), Asp102 (−17.88 kJ/mol), Asp109 (−17.05 kJ/mol), Asp144 (−22.43 kJ/mol), Glu160 (−24.67 kJ/mol), Asp175 (−23.27 kJ/mol), Asp193 (−15.84 kJ/mol), and Asp194 (−32.22 kJ/mol) contributed favorably to nilotinib’s binding (Appendix A). Apart from Met1, all the residues that contributed positive energies were amino acids with positively charged side chains (Arg, His, and Lys), while those that contributed favorable (negative) energies had negatively charged side chains (Glu and Asp). Similar patterns were observed for ZINC000085531689 (Appendix A) and ZINC000014089759 (Appendix A). However, for ZINC000034518176 (Appendix A) and ZINC000095485942 (Appendix A), which were both obtained from the AfroDb library, the opposite pattern was observed. For ZINC000034518176 and ZINC000095485942, amino acid acids with positively charged side chains (except Met and Pro) contributed favorably to ligand binding, while those with negatively charged side chains contributed positive energy values. These residues may be considered key residues and may play major roles in ligand binding.

Regarding ZINC000085568136, only Gln170 and Tyr171 were observed to contribute significantly, presenting energy values of −8.598 and −11.8 kJ/mol, respectively (Appendix A). Pro131 (−5.281 kJ/mol) and Glu160 (7.041 kJ/mol) were observed for ZINC000095912717 (Appendix A), while Thr129 (−5.07 kJ/mol) and Pro131 (−11.29 kJ/mol) contributed significantly toward NANPDB2933 binding (Figure 6). The residues at positions 129–131 and 163–172 contributed favorably to NANPDB2933 binding (Figure 6). Future VP40 drug discovery studies can design compounds with stronger affinity to VP40 by taking these residues into consideration during lead optimization. The chemical mutation of the ligands [168] and the optimization of the partial charges [169,170] of the shortlisted molecules are strategies that can be exploited to improve the binding potency of the compounds in the binding site. Chemical mutation involves the modification of a compound’s chemical structure through the introduction of new functional groups, the substitution of existing ones, or the modification of the compound’s chemical scaffold. Pyridinations, fluorinations, and oxygen-to-sulfur mutations are some common chemical mutations that can be applied to ligands [168] to improve their affinity for specific amino acid residues.

### 2.10. Origin and Sources of the Potential Lead Compounds

Several compound databases, including Zinc15 [171], PubChem [172,173], ChEMBL [174,175,176], LOTUS [177], Indian Medicinal Plants, and Phytochemistry and Therapeutics (IMPPAT) [178], were searched for information on the seven shortlisted potential lead compounds (Table 4). The existing literature on the biological activity of the sources of the compounds was also investigated. NANPDB2933 (2-hydroxyseneganolide), which offers strong antifeedant activity (200 μg/mL and above) [179,180], has been extracted from the fruit [181] and stem bark [179,180] of *Khaya senegalensis*. *K. senegalensis* extracts possess antimicrobial, anti-cancer, and anti-inflammatory activity [182,183]. ZINC000095912717 (lancifodilactone C) is also available in *Schisandra chinensis* (Turcz.) Baill [184] and *Schisandra lancifolia* [185]. Lancifodilactone I to N, isolated from *S. lancifolia*, have previously been shown to possess anti-HIV-1 activity with an EC_50_ ranging from 76.6 to 100.0 μg/mL [186]. Lancifodilactone I and C are structurally similar [186], thus rendering lancifodilactone C an interesting candidate to test experimentally to ascertain its anti-EBOV and antiviral properties.

ZINC000034518176, similar to α-amyrin, is a chemical constituent of *Vernonanthura chamaedrys* [187,188], *Mangifera indica* [189], *Cadia purpurea* [190], *Calendula officinalis* [191], *Pinalia leucantha* [192], *Ilex aquifolium* [193], *I. goshiensis* [194], and *Rhodomyrtus tomentosa* [195] according to Wikidata (https://www.wikidata.org/wiki/Q105000762 (accessed on 15 March 2023)) [196,197]. The organic extract of dried flowers from *Calendula officinalis* exhibits anti-HIV-1 replication properties and is non-toxic to human lymphocytic Molt-4 cells [198]. At 500 μg/mL, this organic extract protected uninfected Molt-4 cells for up to 24 h from infected U-937/HIV-1 cells [198]. ZINC000014089759 (11-keto boswellic acid) has also been extracted from the gum resin and stem bark of *Boswellia papyrifera* [199], *B. sacra* [200,201,202], and *B. serrata* [203,204]. Gum resin extracts of various *Boswellia* spp. have been reported to possess broadly effective antiviral activity [205,206]. 11-keto boswellic acid exhibits anti-inflammatory activity, inhibits 5-lipoxygenase with IC_50_ values of 2.8 to 8.8 μM [207], downregulates TNF-α, and decreases the levels of the proinflammatory cytokines interleukin 1α (IL-1α), IL-2, IL-4, and IL-6 and interferon gamma (IFN-γ) [208,209]. EVD, however, is associated with an intense immune response that leads to the production of proinflammatory cytokines, thereby contributing to the pathogenesis of EVD by causing tissue damage and multiple organ failure [210,211,212]. Inhibiting the expression of proinflammatory cytokines could be a potential therapeutic strategy for EVD. 11-keto boswellic acid was also reported to inhibit prolyl endopeptidase (PEP) with an IC_50_ of 36.32 μM [199].

### 2.11. Limitations of the Study

A major drawback of this study is its lack of experimental testing of the identified compounds to validate their potential anti-VP40 and anti-EBOV activity. Without experimental validation, the effectiveness and safety of these compounds cannot be conclusively established, which limits their potential as viable drug candidates. Therefore, future research efforts should focus on experimental testing to assess the activity and safety of potential lead compounds. Another major caveat of this study concerns the predicted structures from I-TASSER, which was used for molecular docking and dynamics simulations. Since the flexible loop regions in VP40 found between residues 1–43 were not resolved in the parent template (PDB ID: 1ES6), the modeled structure may not accurately reflect the actual structure of VP40. However, the selected I-TASSER model had a C-score of −0.68 and an estimated TM-score of 0.63. I-TASSER-predicted models with TM scores >0.5 and C scores >−1.5 have been reported to have very good false positive and false negative rates of 0.05 and 0.09, respectively [41], implying that the selected structure could be reasonably accurate since it falls within this range. The success of computational drug discovery studies depends on the availability and quality of data. Subjecting the modelled structure to energy minimization and subsequent MD simulations could help fix the structures and provide a reasonably stable starting point for analysis. Furthermore, this study employed multiple robust computational approaches (including the validation of the molecular docking tool) to mitigate any potential errors that could have arisen.

## 3. Materials and Methods

Pharmacoinformatics-based techniques were employed to identify potential VP40 inhibitors from natural-product-derived compounds curated from African and Chinese sources (Figure 7) as performed in a previous study [213]. Approved drugs from various institutions were also obtained and screened (Figure 7). The study used structure-based virtual screening coupled with MD simulations and molecular mechanics Poisson–Boltzmann surface area (MM/PBSA) analyses to propose compounds with plausible binding affinity and mechanisms of binding. The shortlisted compounds were subjected to safety and toxicity profiling to further shortlist those with insignificant toxicity concerns. The biological activity and anti-EBOV activity of the shortlisted compounds were also determined.

### 3.1. Target Retrieval and Preparation

Due to the critical role of the NTD of VP40 in oligomerization and VLP production, this study considered the NTD of the Ebola virus in molecular docking and dynamics simulations. The use of only the NTD did not negatively affect the study as the separation of the CTD from the NTD is required for VP40’s transformation from its dimeric form to its hexameric form or octameric ring structure [214]. The crystal structures of the NTD of the matrix protein VP40 from Zaire Ebola virus with PDB IDs of 1H2C and 7K5L were obtained from the Protein Data Bank (PDB), a global resource that contains a wealth of three-dimensional (3D) information about experimentally determined biological macromolecules [35,215,216]. The 3D structures were visualized using PyMOL (version 2.5.2) [217,218,219]. Due to missing residues, the structure of the NTD of VP40 was remodeled using two different modelling techniques comprising I-TASSER [40,41,42,220] and Modeller (version 10.2) [221,222,223]. The sequence of the VP40 NTD was extracted from UniprotKB since both structures had missing residues. After careful examination of three Zaire EBOV sequences from UniprotKB with IDs of Q05128 (strain Mayinga-76), Q77DJ6 (strain Kikwit-95), and Q2PDK5 (strain Gabon-94), it was observed that all three strains had the same sequence. Using the 7K5L and 3TCQ models as templates, Modeller was employed to generate five structures. The sequence was also submitted to I-TASSER to predict reasonable structures for comparison. The best structure from each technique was selected based on the lowest discrete optimized potential energy (DOPE) score for Modeller and the highest confidence score for I-TASSER. Between the top 2 structures, the model with the most reasonable fold for the missing residues was selected for this study.

Prior to the molecular docking studies, the selected VP40 structure was energy-minimized using GROMACS 5.1.5 [224,225,226] in order to achieve a state of lower potential energy [227,228], thereby rendering the structure more stable. After energy minimization, the resulting structure was visualized in PyMOL, wherein the water molecules and ions were removed.

### 3.2. Binding Site Determination

Existing literature on Ebola VP40 was probed to identify previously determined active sites. The Computed Atlas of Surface Topography of proteins (CASTp) was also used to determine binding sites of the NTD of the VP40 protein [229,230,231].

### 3.3. Obtaining and Preparing Ligand Libraries

A total of 880 small molecules were retrieved from AfroDb, a library of natural compounds of African origin [232]. A total of 6842 compounds were also obtained in 2D spatial data file (sdf) format from the Northern African Natural Products Database (NANPDB) [233] and converted to 3D structures using Open Babel’s “gen3d” option as previously described [213]. NANPDB is a repository of natural products from plant, animal, fungal, and bacterial sources that are available in Northern Africa [233]. A total of 35,161 compounds were also retrieved from the Traditional Chinese Medicine (TCM) database (TCM@Taiwan), a catalog of Zinc15 database [171,234]. Duplicates were removed from these libraries, and the natural products were then pre-filtered based on molecular weight (as described previously) [235]. Only compounds with molecular weights between 150 g/mol and 600 g/mol were used in this study. A total of 773, 3619, and 25,196 compounds from the AfroDb, NANPDB, and TCM libraries had molecular weights within the threshold and were used for this study.

Furthermore, 3D structures of 3094 approved drugs were obtained from Selleckchem Inc. (Houston, TX, USA) and included in the screening library. The approved drugs were curated from various institutions, including U.S. Food and Drug Administration (FDA, Silver Spring, MD, USA), China Food and Drug Administration (CFDA, Beijing, China), European Medicines Agency (EMA, Amsterdam, The Netherlands), Heads of Medicines Agency (HMA, Amsterdam, The Netherlands), and Pharmaceuticals and Medical Devices Agency (PMDA, Chiyoda-ku, Tokyo, Japan).

Literature search was conducted to identify experimentally tested VP40 inhibitors that could be used as standards in this study. Nilotinib and imatinib were shown to reduce the release of VLPs [82]. NP and VP40 levels were also observed to be lower in VLPs treated with nilotinib and imatinib at tested concentrations of 10 and 20 μM [82]. Using a VP40-based VLP assay to screen a library of compounds, sangivamycin (at concentrations of 37.5, 75, 150, and 300 nM) was reported to disrupt EBOV VP40 membrane localization and decrease VLP release; however, it did not affect VP40 abundance [236]. Another study also identified 53 compounds that inhibit entry of VLPs containing the GP and VP40 proteins fused with a beta-lactamase reporter protein [237]. However, the study also highlighted the possibility that some of the drugs were beta-lactamase inhibitors instead of VLP-entry inhibitors [237]. Herein, the 3D structures of these compounds were obtained from PubChem [172,238,239]. Compounds that had no 3D structures were ignored since converting them from 2D could introduce potential errors in their molecular descriptions, such as errors in connectivity, missing bonds, and abnormal geometries, which could influence their docking scores. Due to the uncertainty of whether all 53 compounds were specific Ebola VLP entry inhibitors [237], only nilotinib, imatinib, and sangivamycin were used as standards for this study [82,236]. However, the other inhibitors were also screened against VP40 to investigate their binding to this protein.

The small molecules in “sdf” formats were energy minimized using the universal force field (UFF) [240] via the OpenBabel option in PyRx [241,242]. The compounds were then converted to AutoDock Vina’s compatible “Protein Data Bank, Partial Charge (Q), & Atom Type (T)” (PDBQT) format.

### 3.4. Docking Validation

Prior to the molecular-docking procedure, the ability of AutoDock Vina to discriminate between anti-VP40 actives and inactives was validated using Screening Explorer [59]. The three known inhibitors, comprising nilotinib, imatinib, and sangivamycin, were used to generate 150 decoys (50 each) via Database of Useful Decoys: Enhanced (DUD-E) [243]. The three known inhibitors and the 150 decoys were screened against the VP40 protein, and the results were uploaded to Screening Explorer to compute the area under the receiver operating characteristics curve (ROC AUC), Boltzmann-enhanced discrimination of receiver operating characteristics (BEDROC), total gain (TG), and the robustness initial enhancement (RIE).

### 3.5. Molecular Docking

Virtual screening is a substantive computational approach of tremendous importance in in silico drug design that employs computer-based methods to discover ligands that can bind a protein based on biological structures [244,245]. In this study, AutoDock Vina embedded in PyRx (version 0.9.2) was used to perform molecular-docking processes [242,246]. The grid box was defined to cover the RNA binding site and the loop region with dimensions of (39.05 × 32.15 × 33.42) Å^3^ and centered at 30.17, 27.82, and 38.36 Å. An exhaustiveness value of eight was also set for the docking run. After docking, the ligands were ranked in descending order based on their docking scores. A good docking score does not imply a strong binding affinity as previous studies have shown that docking scores are not reliable in terms of ranking the binding potency of compounds [247,248,249]. Nevertheless, molecular docking has been shown to reasonably generate conformations that are similar to experimentally determined protein–ligand complex structures [247,248]. Thus, a good docking score provides a conformation/pose, which can be used as a good starting point for MD simulation-based analyses [247]. Herein, compounds with docking scores lower than that of the best-performing inhibitor were shortlisted for further analysis.

### 3.6. Pharmacological and Toxicity Profiling of Molecules

The drug likeness of the shortlisted compounds was assessed using Lipinski’s rule of 5 and Veber’s rule. ADMET properties usually depend on the molecular descriptors of the compound. The possibility for the in silico prediction of these properties for new structures allows one to considerably accelerate drug design and optimization to increase efficiency. ADMET constitutes the structural, physiochemical, biochemical, and pharmacokinetic properties of a drug molecule as well as its toxicity, which are essential for evaluating its pharmacodynamics. The interactions of the structural properties of a compound with its physical environment can govern its physiochemical properties (e.g., solubility, lipophilicity, stability, etc.), while interactions with proteins are important determinants of its biochemical properties (e.g., metabolism). At the highest level, when these physicochemical and biochemical properties interact with living systems, they confer pharmacokinetic effects and toxicity [250]. The pharmacological properties of the compounds were evaluated using SwissADME [251]. DataWarrior 5.5.0 was also used to predict the toxicity risks of the shortlisted compounds [89].

### 3.7. Protein–Ligand Interaction

In addition to the binding affinity of the molecules, the inhibitory effect of compounds can be determined by analyzing their interactions with receptor molecules. In particular, multiple hydrogen bond interactions ensure the stability of drug–receptor complexes and may influence the activity of the drug [163]. LigPlot^+^ (v1.4.5) and PyMOL were used to visualize the binding interactions between the VP40 and the top compounds.

### 3.8. Prediction of Biological Activities of Lead Compounds

The biological activities of the identified lead compounds were predicted using prediction of activity spectra for substances (PASS) [90]. In this method, prediction is based on an analysis of the structure–activity relationships in the training set containing information on the structure and biological activity of more than 300,000 compounds. PASS may be used to find new targets for known pharmaceuticals or for searching for new biologically active substances. Compounds in the Simplified Molecular Input Line Entry System (SMILES) format were used as inputs. In addition, anti-Ebola, a regression-based algorithm, was employed to predict the potential EBOV-inhibitory activities of the selected compounds using their SDF formats. Anti-Ebola predicts the inhibition efficiency of compounds using quantitative structure–activity relationship (QSAR) information of molecules that have experimentally tested anti-Ebola activities [94]. The random forest (RF) and support vector machine (SVM) algorithms of Anti-Ebola were employed for the activity predictions conducted in this study [94].

### 3.9. MD Simulations of VP40 and VP40–Ligand Complexes

MD simulations were performed on the unbound VP40 and VP40–ligand complexes using GROMACS 5.1.5 [224]. Ligand topologies were generated using LigParGen [252]. The all-atom optimized potentials for liquid simulations (OPLS/AA) force field was used to prepare the topology of the VP40 protein [253,254]. Each system was solvated in a cubically shaped box and neutralized by the addition of chlorine or sodium ions depending on the system’s charge. Energy minimization was performed for each system for 1000 steps using the steepest descent algorithm and then further equilibrated to restrain and relax protein and ligand positions. Finally, an MD simulation of the complexes was performed for 100 ns, and Qtgrace was used to plot all graphs generated from the MD simulations [255].

### 3.10. Evaluation of Lead Compounds Using MM/PBSA

MM/PBSA calculations were performed for the VP40–ligand complexes after the 100 ns MD simulation using g_mmpbsa [164]. The *van der Waals* (vdW), electrostatic, polar solvation, solvent-accessible surface area (SASA), and binding free energies of each complex were computed. The per-residue energy contributions of each complex were also determined to evaluate the critical residues favoring or preventing ligand binding. The R programming package was used to plot the graphs of free binding energy.

## 4. Conclusions

Over the years, computer-aided drug design (CADD) techniques have proven to be advantageous for the discovery of effective treatments for different diseases. Therefore, this study employed similar techniques to identify potential anti-EBOV VP40 lead compounds from natural product sources and to repurpose currently approved drugs. This study bolsters the findings of existing studies undertaken to find potential EBOV inhibitors to increase our therapeutic arsenal against EVD [94,237,256,257,258]. A total of 32,300 ligands were successfully screened against VP40, of which 65 (42 natural products and 23 approved drugs) were predicted to be potential anti-EBOV compounds. The potential lead compounds demonstrated better docking scores with respect to VP40 (−8 to −9.1 kcal/mol) than the three known inhibitors (nilotinib, imatinib, and sangivamycin) used (−6.3 to −7.9 kcal/mol). MD simulations and MM/PBSA computations also showed that the potential lead compounds had stronger binding affinity to VP40 (−46.97 to −118.9 kJ/mol) than nilotinib (−11.21 kJ/mol), which was used as a control. The in silico characterization of the potential lead compounds using ADMET testing corroborated their drug-like properties. It was observed that most of the compounds bind in the loop region rather than the RNA binding site as suggested previously [48]. The biological activities of the lead compounds that were predicted using PASS suggest that these lead compounds possess potential anti-viral mechanisms of action for inhibiting viral replication and budding. The promising potential lead compounds identified herein have the propensity to act as anti-Ebola VP40 scaffolds. The results obtained indicate that these compounds deserve to be experimentally tested in vitro and in vivo to ascertain their ability to combat the infection and spread of EVD.

## Figures and Tables

**Figure 1 ijms-24-06298-f001:**
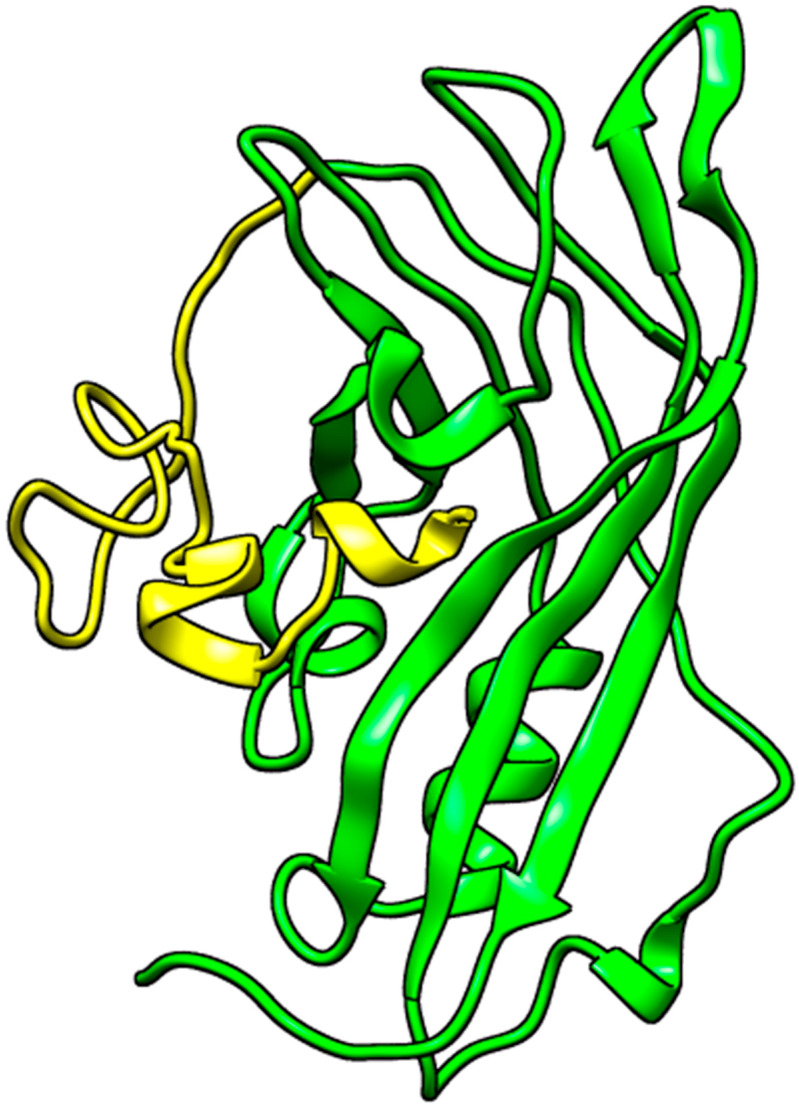
Structure of the selected VP40 NTD model generated via I-TASSER using 1ES6’s structure as template. Image was generated using UCSF Chimera version 1.16 [45]. Regions colored yellow represent missing residues that were remodeled.

**Figure 2 ijms-24-06298-f002:**
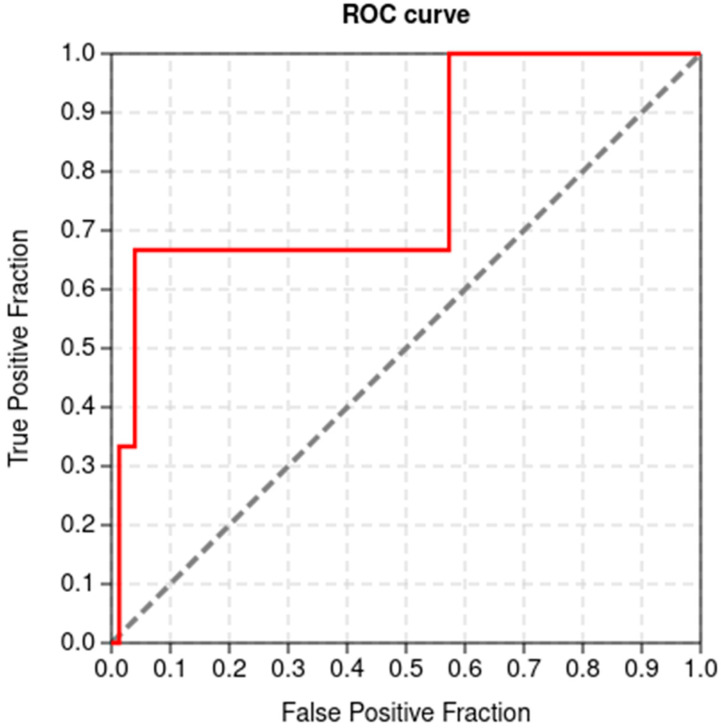
ROC curve generated using 150 decoys and 3 known inhibitors of EBOV VP40 to assess molecular docking performance. An AUC ROC value of 0.791 was obtained.

**Figure 3 ijms-24-06298-f003:**
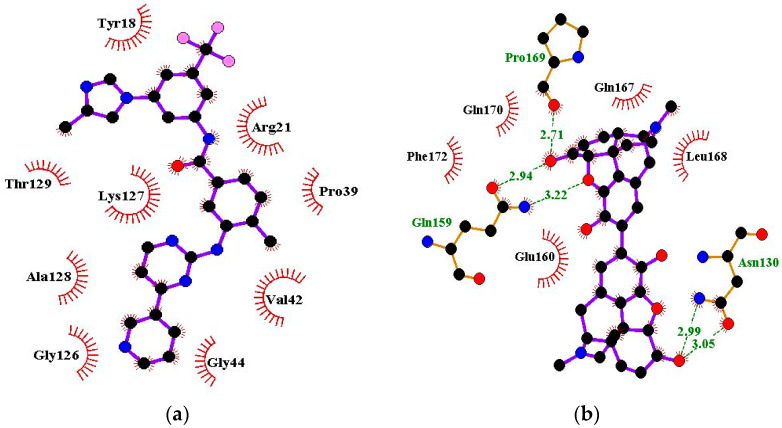
Protein–ligand interaction map of VP40 in complex with (**a**) nilotinib and (**b**) ZINC000085531689. The green, dotted lines indicate hydrogen bonds while the red arcs with spikes represent hydrophobic interactions.

**Figure 4 ijms-24-06298-f004:**
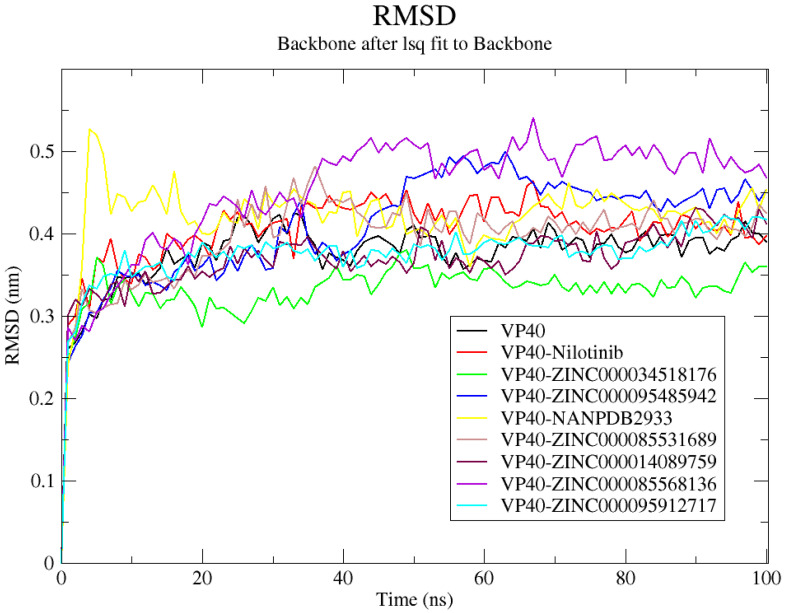
RMSD plot of the EBOV VP40 and VP40–ligand complexes after 100 ns MD simulations. The unbound VP40 protein, VP40–nilotinib, VP40–ZINC000034518176, VP40–ZINC000095485942, VP40–NANPDB2933, VP40–ZINC000085531689, VP40–ZINC000014089759, VP40–ZINC000085568136, and VP40–ZINC000095912717 complexes are colored black, red, green, blue, yellow, brown, maroon, violet, and cyan, respectively.

**Figure 5 ijms-24-06298-f005:**
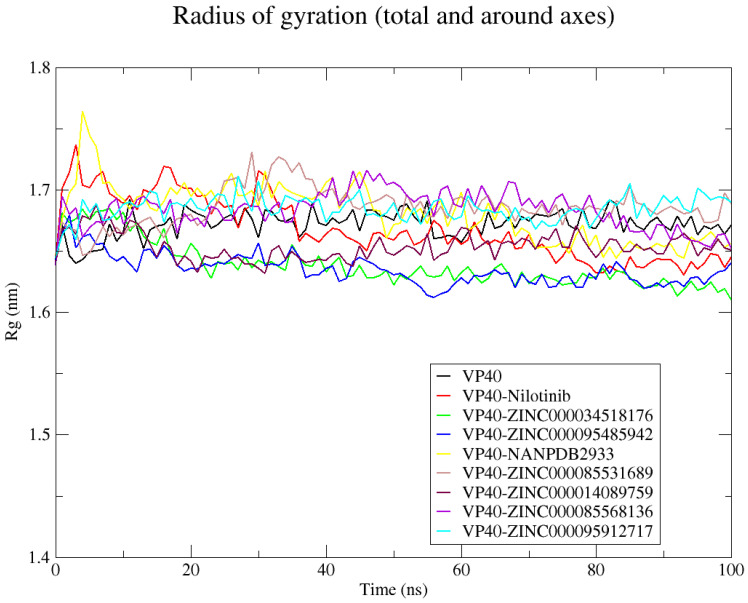
Rg plot of the EBOV VP40 and VP40–ligand complexes after 100 ns MD simulations. The unbound VP40 protein, VP40–nilotinib, VP40–ZINC000034518176, VP40–ZINC000095485942, VP40–NANPDB2933, VP40–ZINC000085531689, VP40–ZINC000014089759, VP40–ZINC000085568136, and VP40–ZINC000095912717 complexes are colored black, red, green, blue, yellow, brown, maroon, violet, and cyan, respectively.

**Figure 6 ijms-24-06298-f006:**
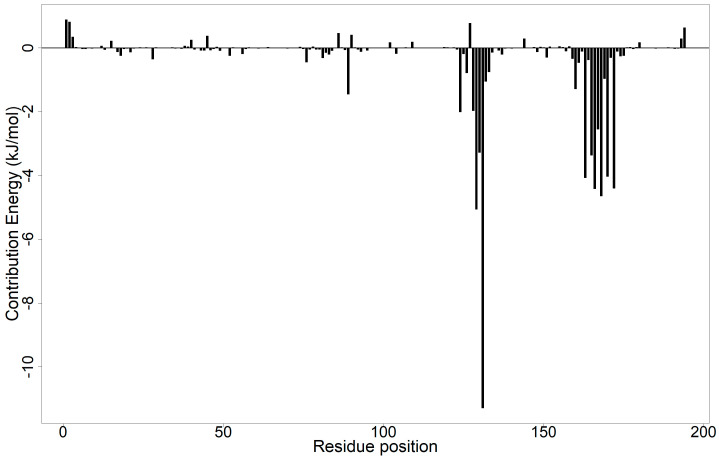
Per-residue energy decomposition plot of the VP40–NANPDB2933 complex showing the energy contributions of each amino acid residue.

**Figure 7 ijms-24-06298-f007:**
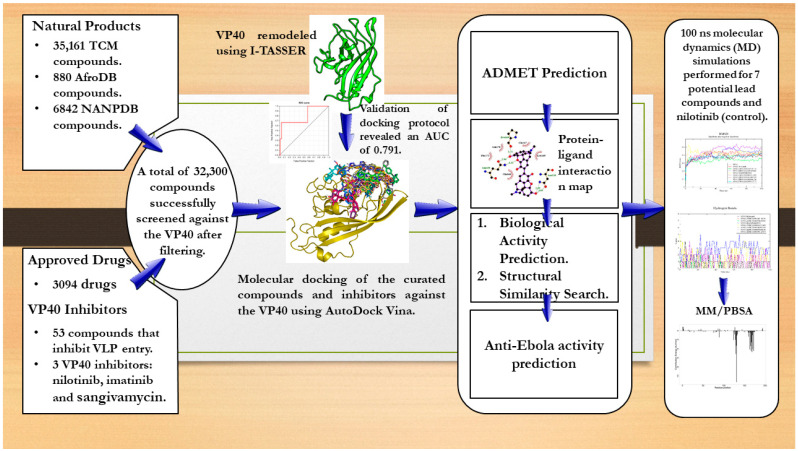
Schema of methodology detailing the step-by-step approach used to identify potential anti-VP40 compounds.

**Table 1 ijms-24-06298-t001:** Docking scores and interacting residues of some top compounds that were docked into VP40’s NTD.

Compound	Source/Library	Docking Score	Hydrogen Bonds (Bond Length (Å))	Hydrophobic Contacts
ZINC000034518176	AfroDb	−8.1	Gly139 (3.05)	Phe36, Asn43, Pro47, Thr121, Phe125, Arg134, Asn136, Arg137, and Leu138.
ZINC000095485942	AfroDb	−8.1	Thr129 (2.72) and Asn130 (2.85, 3.25)	Pro131, Gln159, Glu160, Pro165, Val166, Pro169, and Gln170.
NANPDB2933	NANPDB	−8.5	Val166 (3.23) and Gln167 (3.09, 3.26).	Ala128, Thr129, Pro131, Gln159, Glu160, Pro165, and Gln170.
ZINC000085531689	TCM	−8.9	Asn130 (2.99, 3.05), Gln159 (2.94, 3.22), and Pro169 (2.71).	Glu160, Gln167, Leu168, Gln170, and Phe172.
ZINC000014089759	TCM	−8.8	Arg21 (3.02, 3.03), Pro39 (3.1), and Lys127 (3.04).	Tyr13, Tyr18, Pro19, and Asn23.
ZINC000085545967	TCM	−8.8	Gln159 (3.16), Leu168 (3.14), and Leu169 (3.16).	Lys127, Ala128, Thr129, Pro131, Glu160, Pro165, Val166, Gln167, Gln170, and Phe172.
ZINC000085568136	TCM	−8.7	Asp45 (3.11) and Gly84 (3.13).	Val42, Gly44, Gly126, Lys127, Als128, Tyr171, and Thr173.
ZINC000095912717	TCM	−8.7	Gln159 (3.09), Val166 (3.14), and Gln170 (3.04).	Ala128, Asn130, Pro131, Glu160, Pro164, Pro165, Gln167, Leu168, and Pro169.
ZINC000014089743	TCM	−8.6	Arg21 (2.96), Ser24 (2.97), and Pro39 (3.09).	Tyr13, Tyr18, Pro19, Asn23, and Lys127.
ZINC000101564200	TCM	−8.6	Val166 (2.78, 3.09), Pro169 (2.77), and Gln170 (3.09).	Lys127, Ala128, Thr129, Pro131, Gln159, Glu160, Pro165, Gln167, and Leu168.
ZINC000085504890	TCM	−8.5	Asn130 (3.23) and Gln170 (3.16)	Lys127, Ala128, Thr129, Pro131, Gln159, Glu160, Pro165, Val166, Gln167, Leu168, and Pro169.
ZINC000095909661	TCM	−8.5	Asn130 (2.8) and Pro169 (3.18)	Ala128, Thr129, Pro131, Gln159, Glu160, Leu168, and Gln170.
ZINC000070454124	TCM	−8.4	Val42 (3.05) and Thr129 (3.19).	Val20, Arg21, Pro39, Asn43, Gly44, Lys127, Ala128, Leu132, and Tyr171.
Doramectin	Approved	−9.1	Glu12 (2.72), Asn130 (3.14), Gln159 (2.8), Leu168 (3.33), and Pro169 (3.06).	Tyr13, Lys127, Ala128, Thr129, Pro131, Glu160, Phe161, Pro165, and Gln170.
Ledipasvir	Approved	−9	-	Arg21, Pro39, Val42, Asn43, Gly44, Asp45, Thr46, Ser83, Gly84, Lys127, Ala128, Thr129, and Tyr171.
Avermectin B1 (Abamectin)	Approved	−8.7	Glu12 (2.89), Asn130 (3.04), Gln159 (2.85), and Leu168 (3.11)	Thr8, Tyr13, Ala128, Thr129, Pro131, Glu160, Phe161, Pro165, Pro169, Gln170, and Phe172.
Elbasvir	Approved	−8.7	Asp45 (3.16)	Arg21, Pro39, Val42, Asn43, Gly44, Thr46, Ser48, Gly84, Lys127, Ala128, Thr129, Tyr171, Thr173, Phe174, and Asp175.
Venetoclax (ABT-199, GDC-0199)	Approved	−8.5	Gly44 (3.16), Thr46 (2.88), Gly84 (3.07), and Thr173 (2.85).	Ser48, Asn49, Ile82, Ser83, Leu168, Pro169, Gln170, Tyr171, and Asp175.
Revefenacin	Approved	−8.5	Asp45 (2.9) and Lys127 (2.8)	Tyr18, Val20, Arg21, Pro39, Val42, Gly44, Ser83, Ala128, Thr129, Tyr171, and Phe172.
Glecaprevir	Approved	−8.4	Asn130 (2.93), Gln159 (2.99), Val166 (2.8), and Leu168 (3.31).	Ala128, Thr129, Glu160, Phe161, Pro165, Gln167, Pro169, and Gln170.
Nilotinib	Inhibitor	−7.9	-	Tyr18, Arg21, Pro39, Val42, Gly44, Gly126, Lys127, Ala128, Thr129.
Imatinib	Inhibitor	−7.6	-	Pro39, Glu40, Ser41, Val42, Gly44, Asp45, Ser83, Gly84, Gly126, Lys127, and Tyr171.
Cepharanthine	Inhibitor	−7.3	-	Asn130, Pro131, Gln159, Glu160, Pro165, Val166, Gln167, and Leu168.
Sangivamycin	Inhibitor	−6.3	Pro169 (3.03, 3.2) and Val166 (2.82, 3.11).	Ala128, Thr129, Pro131, Gln159, Glu160, Pro165, Leu168, Gln170, and Phe172.

**Table 2 ijms-24-06298-t002:** Physicochemical and biological properties of some of the shortlisted compounds predicted using SwissADME.

Compound	Docking Score (kcal/mol)	Molecular Weight (g/mol)	TPSA (Å^2^)	LogP	ESOL Solubility Class	GI Absorption	BBB Permeant	Pgp Substrate
**AfroDb**								
ZINC000034518176	−8.1	426.72	20.23	7.03	Poorly soluble	Low	No	No
ZINC000095485942	−8.1	474.5	135.8	1.4	Soluble	High	No	Yes
**NANPDB**								
NANPDB2933	−8.5	486.51	132.5	1.6	Soluble	High	No	Yes
**TCM**								
ZINC000085531689	−8.9	568.66	105.86	2.57	Moderately soluble	High	No	Yes
ZINC000014089759	−8.8	470.68	74.6	5.36	Poorly soluble	High	No	Yes
ZINC000085545967	−8.8	580.79	118.22	4.14	Poorly soluble	High	No	Yes
ZINC000085568136	−8.7	598.66	100.74	4.98	Poorly soluble	High	No	Yes
ZINC000095912717	−8.7	544.59	134.66	1.59	Soluble	High	No	Yes
ZINC000014089743	−8.6	456.7	57.53	6.12	Poorly soluble	Low	No	No
ZINC000101564200	−8.6	478.49	115.06	4.69	Poorly soluble	Low	No	No
ZINC000085504890	−8.5	536.66	101.16	1.97	Moderately soluble	High	No	Yes
ZINC000095909661	−8.5	594.7	83.86	4.75	Poorly soluble	High	No	No
ZINC000070454124	−8.4	564.58	119.34	3.83	Moderately soluble	High	No	Yes
ZINC000085530485	−8.4	566.64	124.8	3.15	Moderately soluble	High	No	Yes
ZINC000095911418	−8.4	462.62	52.6	5.63	Moderately soluble	High	No	No

**Table 3 ijms-24-06298-t003:** Contributing energies of the VP40–ligand complexes estimated from MM/PBSA computation. Values are presented as average energy value ± standard deviation in kJ/mol.

Compound	vdW	Electrostatic	Polar Solvation	SASA	Binding
ZINC000034518176	−117 ± 1.444	−33.3 ± 2.86	60.3 ± 2.132	−15.15 ± 0.164	−105 ± 1.815
ZINC000095485942	−93.49 ± 2.464	−80.21 ± 5.726	82.09 ± 6.387	−11.64 ± 0.257	−103.3 ± 2.217
NANPDB2933	−166.5 ± 1.572	−50.07 ± 1.021	114.5 ± 1.085	−16.84 ± 0.098	−118.9 ± 1.838
ZINC000085531689	−108.9 ± 2.797	−47.25 ± 4.294	123.2 ± 5.171	−14.06 ± 0.333	−46.97 ± 3.062
ZINC000014089759	−97.39 ± 3.274	15.76 ± 2.801	23.02 ± 1.793	−12.08 ± 0.395	−70.52 ± 2.343
ZINC000085568136	−156.5 ± 2.969	−28.37 ± 1.995	110.9 ± 2.643	−18.6 ± 0.306	−92.71 ± 2.553
ZINC000095912717	−131.9 ± 1.218	−21.38 ± 1.359	101.9 ± 1.817	−15.76 ± 0.147	−66.99 ± 1.73
Nilotinib	−108.5 ± 3.918	41.83 ± 5.391	71.17 ± 4.661	−15.75 ± 0.56	−11.21 ± 3.55

**Table 4 ijms-24-06298-t004:** Names, PubChem IDs, 2D structures, and sources of the potential lead compounds. Marvin 23.3.0 by ChemAxon (https://www.chemaxon.com accessed 11th March, 2023) was used to generate the 2D structures.

Compound ID (PubChem Compound ID)	Common Name/IUPAC Name	Source/Origin	2D Structure
ZINC000034518176 (CID: 10836206)	(3S,4aR,6aR,6bS,8aR,11R,12S,12aS,14aR,14bR)-4,4,6a,6b,8a,11,12,14b-octamethyl-2,3,4a,5,6,7,8,9,10,11,12,12a,14,14a-tetradecahydro-1H-picen-3-ol	Vernonanthura chamaedrys [187,188], *Mangifera indica* [189], *Cadia purpurea* [190], *Calendula officinalis* [191], *Pinalia leucantha* [192], *Ilex aquifolium* [193], *I. goshiensis* [194], and *Rhodomyrtus tomentosa* [195]	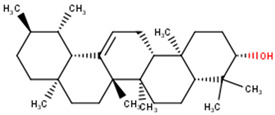
ZINC000095485942 (CID: 163021364)	(1*R*,2*R*,4*R*,7*S*,8*R*,10*R*,11*S*,12*S*,13*R*,18*R*)-7-(furan-3-yl)-10,13-dihydroxy-8,12,17,17-tetramethyl-3,6,16-trioxapentacyclo [9.9.0.0^2,4^.0^2,8^.0^12,18^]icosane-5,15,20-trione	-	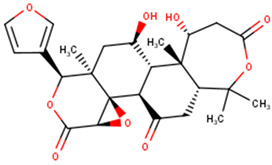
NANPDB2933 (CID: 102019659)	2-hydroxyseneganolide	*Khaya senegalensis* [179,180,181]	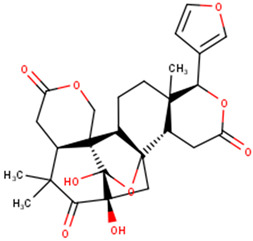
ZINC000085531689 (CID: 97042015)	(4*S*,4*aR*,7*S*,7*aR*,12*bR*)-10-[(4*S*,4*aS*,7*R*,7*aS*,12*bR*)-7,9-dihydroxy-3-methyl-2,4,4*a*,7,7*a*,13-hexahydro-1*H*-4,12-methanobenzofuro [3,2-e]isoquinolin-10-yl]-3-methyl-2,4,4*a*,7,7*a*,13-hexahydro-1*H*-4,12-methanobenzofuro [3,2-e]isoquinoline-7,9-diol	-	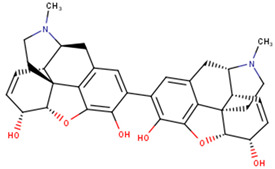
ZINC000014089759 (CID: 9847548)	11-keto boswellic acid	*Boswellia papyrifera* [199], *Boswellia sacra* [200,201,202], and *Boswellia serrata* [203,204]	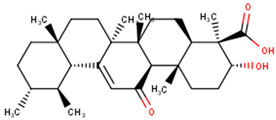
ZINC000085568136 (CID: Not available)	5-[(11R,12R,17R,18S)-12,22-dihydroxy-11-(4-hydroxy-3-methoxyphenyl)-4,10-dioxahexacyclo [15.7.1.0^2,15^.0^3,8^.0^9,14^.0^21,25^]pentacosa-1(25),2,8,14,19,21,23-heptaen-18-yl]-2H-indol-2-ylium	-	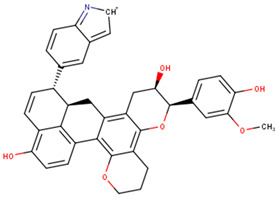
ZINC000095912717 (CID: 12080815)	Lancifodilactone C	*Schisandra chinensis* (Turcz.) Baill [184] and *Schisandra lancifolia* [185]	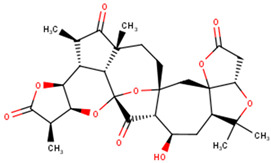

## Data Availability

Not applicable.

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
