# Peer review of "Cheminformatics-Based Study Identifies Potential Ebola VP40 Inhibitors"

_ijms, 2023, doi:10.3390/ijms24076298_

Round 1

Reviewer 1 Report

It is difficult for me to assess the extent to which research conducted exclusively by in silico is consistent with the goals of the IJMS. I believe that since the Editor allowed this manuscript to be considered by reviewers, then it corresponds. However, the manuscript in this form raises questions for me.

In the Introduction, the authors made a completely incorrect statement "Natural products as well demonstrate less toxicity compared to synthetic products" (Lines 115-116), and even supported by two links, this sentence is completely unprofessional. This needs to be corrected.

The conclusions of the authors in this manuscript are not confirmed by any experimental data, therefore, the authors need to provide at least literary data on the biological activity of those compounds that they discovered as possible leader molecules. The authors need to give the structures of leader molecules, their origin and activity in Discussion section. And at least in this way to verify your results.

Reviewer 2 Report

The paper presented in silico docking study of natural product compounds from different libraries and approved drugs for Ebola virus VP40 binding. As a result, 42 natural product compounds and 23 approved drugs were identified to have the potential to bind and inhibit the Ebola virus VP40.

One of the main databases in this study was used in a previously published paper (Titled: Virtual screening of the inhibitors targeting the viral protein 40 of Ebola virus) for the same screening purpose. Please explain the differences between the studies and the advancement from this study. Though it was claimed in the study that the lead compounds demonstrated stronger binding in silico than previously identified molecules, none of the hits were verified by further experiments. The methodology used in this study was also previously published (For example, https://doi.org/10.3390/biom11030458) With all previous studies and identified molecules for potential inhibition, there is little significance in adding in just more molecules. 

Abstract: please list the full name prior to using the acronym. The abstract can be shortened to be more concise.

Line 160: please explain what you mean by reasonably best or the criteria are.

Lien 161: please cite the corresponding source for GROMACS

Line 180: please check and rewrite. The sentence does not make sense. Quote “Only compounds with molecular weights between 150 g/mol and above 600 g/mol were used in this study.”

Redundancy in the result sections. Some content can be moved to method sections or duplicated from the method sections.

Reviewer 3 Report

In this work, the authors performed a series of computational modeling to screen ~30000 compounds to indentify potnetial Ebola virus inhibitors. I hope the authors can address my comments below:

1. The authors need to pay more attention to significant figures in their reported data. I can see both redundant and inconsistent significant figures. The authors need to check their manuscript thoroughly to correct them. 

2. In Figure 2, I suggest the authors use different colors to highlight the residues that were missing and rebuilt.

3. Section 3.2: is there any experimental data (HDX, NMR, etc.) available to support the predicted binding site? This is the foundation of the work so I hope the authors can provide more evidence to show that the predicted binding site is reliable. 

4. Section 3.4: I notice the authors consider docking scores as binding affinities. In fact, this is an incorrect interpretation of docking scores. A docking score is not equivalent to a binding affinity. The docking score can be used to rank docked poses but does not indicate how tight the compound can bind to the receptor. Many papers have proved that using docking scores to rank the compounds for their binding potency is not reliable. (see https://doi.org/10.1002/cmdc.202200425 and https://doi.org/10.1021/jm050362n) The authors should clarify this in the manuscript when talking about docking scores. The authors should also highlight this fact with these previous studies (or even more related work) to make sure readers do not mis-use docking scores in their own research. In general, docking is good for predicting binding poses and providing a starting point for molecular dynamics simulations. But docking is not an ideal tool to predict binding free energies and rank compounds for their binding potency. Also the authors need to check the manuscript thoroughly to make sure the "Binding Energy" or "Binding Affinity" are replaced by "Docking Score" and correct any statements based on docking scores. 

5. Section 3.4: The authors used a threshold of -8.0 kcal/mol based on previous work using AutoDock Vina. While the docking score is only meaningful for the same series of compounds screened against the same receptor. It is not transferable between receptors or compounds to screen. The authors should determine a threshold based on the obtained scores themselves but not from another study on a different target or compound library. 

6. Section 3.6: It is more appropriate to analyze the protein-ligand interactions from molecular dynamics (MD) simulations. Docking poses need to be verified by MD simulations and the interactions can be better understood from simulations than docking poses. 

7. Line 521: Are these studies based on experimental results? If so, the authors should highlight it. The work is purely based on computational predictions so any experimental validation is helpful to support the predictions. 

8. Section 3.7.2: It is unclear to me about the goal of performing the structural similarity search. I suggest the authors to clarify more in the manuscript. 

9. Section 3.8.2: I understand the RMSD analysis in Section 3.8.1 is useful to check the convergence. But it is unclear to me in terms of the purpose of analyzing RMSF in this section. What can we learn from this analysis, why it is important and how we can use the information from this analysis?

10. Line 743: is the binding free energy measured from experiment? Or is it from MMPBSA calculation?

11. Line 760: I understand the energy contribution < -5.0 kJ/mol may be important but why the contribution > +5.0 kJ/mol is important? Is that because those residues impair the binding potency?

12. Line 789: Can the authors clarify how the future studies can use these information in lead optimization?

Round 2

Reviewer 1 Report

The revised manuscript can be published.

Reviewer 2 Report

One major caveat of the study based on a computationally generated model is that programs like i-Tasser produce the model based on known structures. Since the flexible loops in the previous structure (which is what the computation is based on ) were not resolved, the computational model may not reflect the actual structure. The intrinsic nature of those loops is flexibility. The confidence of the loop region correctly represented by the model is low. Please consider including more limitation discussions. 

Reviewer 3 Report

The authors addressed my comments in the revised manuscript.
